# Quantitative Investigation of *FAD2* Cosuppression Reveals RDR6-Dependent and RDR6-Independent Gene Silencing Pathways

**DOI:** 10.3390/ijms242417165

**Published:** 2023-12-06

**Authors:** Yangyang Chen, Hangkai Ku, Yingdong Zhao, Chang Du, Meng Zhang

**Affiliations:** College of Agronomy, Northwest A&F University, Yangling 712100, China; cyy2017050121@nwsuaf.edu.cn (Y.C.); khk@nwsuaf.edu.cn (H.K.); zhaoyd@nwafu.edu.cn (Y.Z.)

**Keywords:** cosuppression, gene silencing model, RNA decay, *Arabidopsis*

## Abstract

The frequency and extent of transgene-mediated cosuppression varies substantially among plant genes. However, the underlying mechanisms leading to strong cosuppression have received little attention. In previous studies, we showed that the expression of *FAD2* in the seeds of *Arabidopsis* results in strong RDR6-mediated cosuppression, where both endogenous and transgenic *FAD2* were silenced. Here, the *FAD2* strong cosuppression system was quantitatively investigated to identify the genetic factors by the expression of *FAD2* in their mutants. The involvement of DCL2, DCL4, AGO1, and EIN5 was first confirmed in *FAD2* cosuppression. SKI2, a remover of 3′ end aberrant RNAs, was newly identified as being involved in the cosuppression, while DCL3 was identified as antagonistic to DCL2 and DCL3. *FAD2* cosuppression was markedly reduced in *dcl2*, *dcl4*, and *ago1*. The existence of an RDR6-independent cosuppression was revealed for the first time, which was demonstrated by weak gene silencing in *rdr6 ein5 ski2*. Further investigation of *FAD2* cosuppression may unveil unknown genetic factor(s).

## 1. Introduction

In early plant transgenic research, failure to achieve a predicted effect was often attributed to unexpected technical or methodological difficulties. Later, it was found that transformation with an exogenous gene suppressed the expression of an endogenous gene copy [1]. Subsequently, two groups simultaneously reported that all-white flowers or flowers with white segments appeared when sense *chalcone synthease* (*CHS*) was introduced into pigmented petunia petals to improve pigments in flowers [2,3]. Because decreased expression levels of both endogenous and transgenic CHS were observed, this transgenic gene silencing was named cosuppression [3]. Gene silencing can occur either at transcriptional (transcriptional gene silencing, TGS) or post-transcriptional (post-transcription gene silencing, PTGS) levels. Cosuppression was shown to be PTGS [4] and was also referred to as sense transgene-induced PTGS (or S-PTGS). In most cases, cosuppression happens at a low frequency, such as in the overexpression of *Brassica napus Fatty acid desaturase 3* (*BnFAD3*, encoding a cytochrome b5-dependent linoeoyl omega 3 desaturase) with about 7% cosuppression of the endogenous *FAD3* in *Arabidopsis* [5]. It is believed that some factors, such as tandemly linked, inversely repeated, or methylation-modified transgenes, incidentally trigger gene silencing [6]. However, some cosuppression phenomena occur at a high frequency. The frequency of *CHS* cosuppression in the example cited above was 60–80%. Frequencies of *agonaute 1* (*AGO1*) [7], *chlorophyll synthase* (*CHLSYN*) [8], and *nitrate reductase 2* (*NIA2*) [9,10] cosuppression were 90%, 90%, and 100%, respectively. Cosuppression triggered by different genes shows various silencing frequencies [11]. Due to the limited number of documented cases of strong cosuppression, the specific mechanisms by which this strong silencing is initiated or maintained are not yet well understood.

Transgenic gene silencing may be similar to host defense responses following transposon or viral infection, and they may share a similar mechanism in plants [6]. Genetic factors involved in gene silencing have been identified with forward and reverse genetic approaches. PTGS begins with RNA-dependent RNA polymerase 6 (RDR6)-mediated production of double-strand RNAs (dsRNAs) [12,13] that are cut into small interference RNAs (siRNAs) with DICER-LIKE (DCL) proteins, which are members of the RNase III family [14]. There are four DCLs (DCL1-DCL4) in *Arabidopsis*, and they produce siRNAs of varying lengths [15,16]. siRNAs then bind the effector protein ARGONAUTE (AGO) that guides a sequence-specific degradation of mRNAs (Béclin et al., 2002). Cosuppression of *AGO1*, *CHLSYN,* and *NIA2* has also been shown to be RDR6-dependent [7,8,10]. DCL2 and DCL4 function redundantly in the cosuppression of *AGO1* [7]. More examples are needed to determine whether all strong cosuppressions share a similar mechanism.

Aberrant RNA (abRNA) can be degraded with an RNA decay system or lead to gene silencing by RDRs [17,18]. abRNAs may be degraded in a 5′-3′ or 3′-5′ direction with XRNs (nuclear exoribonucleases) or SKIs (RNA helicase subunits of the SKI complex), respectively [19]. Endogenous PTGS occurs widely in the *ein5-1* (*ethylene insensitive 5*/*xrn4*) *ski2-3* double mutant [20]. abRNA is considered to be a trigger of transgene silencing [21]. Transgenic gene silencing is enhanced by mutations of XRNs [22,23], suggesting that abRNA with an aberrant 5′ end is involved. Moreover, the absence of either SKI2 or SKI3 may provoke the transition from the non-silenced transgene state to PTGS [20,24]. In a study of *β-glucuronidase* (*GUS*) silencing (there is no homologous gene in *Arabidopsis*), *ski3* was less efficient than the *xrn4*/*ein5* mutant in enhancing silencing [24]. It is unclear whether SKIs and cytoplasmic XRN also affect a strong cosuppression. 

Fatty acid desaturase 2 (FAD2) catalyzes the desaturation of oleic acid (18:1) to produce linoleic acid (18:2), which is further desaturated by FAD3 to form α-linolenic acid (18:3) [25,26]. In our previous studies, strong cosuppression was observed in more than 80% of transgenic lines when *FAD2* was overexpressed in flax (*Linum usitatissimum*), *Camelia sativa*, and *Brassica carinata*. In the *Arabidopsis*
*rdr6* mutant, strong cosuppression was released [27]. 

This study is aimed at revealing possible genetic factors involved in S-PTGS using the *FAD2* cosuppression system. Here, further investigation of *FAD2* cosuppression in *Arabidopsis* revealed that DCL2, DCL4, and AGO1 are also mediators in this cosuppression, which is consistent with previous studies of other systems. Moreover, DCL3, XRN4/EIN5, and SKI2 have a negative effect on *FAD2* cosuppression. Additionally, for the first time, we show the presence of an RDR6-independent cosuppression in the *rdr6-11 ein5-1 ski2-3* triple mutant. Our results suggest that the strength of *FAD2* cosuppression may merit adoption as a new model system to further investigate the initiation, maintenance, and genetic factors involved in the strong cosuppression phenomenon. Understanding the mechanism underlying *FAD2* cosuppression may assist in designing strategies for improving 18:2 in oilseeds and may have more general applications in overcoming the cosuppression of other genes.

## 2. Results

### 2.1. DCL2 and DCL4 Function Redundantly and AGO1 Mediates in FAD2 Cosuppression

In *Arabidopsis*, four DCLs have been identified [16], and DCL1 is involved in the generation of miRNAs [28,29]. Considering that DCL2, DCL3, and DCL4 may play roles in *FAD2* cosuppression, the same *FAD2* construct (Pha::*FAD2*) used in a previous study [5], driven by the Phaseolin promoter, was transferred into *dcl2-1*, *dcl3-1*, and *dcl4-2* mutant backgrounds. T_1_ seeds of each transformation experiment were collected, and fatty acid profiles were determined. It was found that the (18:2 + 18:3)/18:1 ratio decreased in most Pha::*FAD2*/*dcl2-1* and Pha::*FAD2*/*dcl4-2* lines, which is similar to that found in Pha::*FAD2*/Col-0 (Figure 1A–D), suggesting that *FAD2* cosuppression is not released in the single mutants of DCL2 and DCL4 (Figure 1A,B,D and Appendix A). It is noteworthy that the cosuppression frequency was slightly higher in the *dcl3-1* mutant than in the WT (Figure 1A,C and Appendix A). Moreover, the reduction in the (18:2 + 18:3)/18:1 ratio in T_1_ seeds of Pha::*FAD2*/*dcl3-1* lines was more marked than that observed in the WT transgenic lines (Figure 1A,C and Appendix A), indicating that *FAD2* cosuppression is enhanced in the *dcl3-1* mutant.

To test for possible additive or redundant effects of these DCLs on *FAD2* cosuppression, the Pha::*FAD2* constructs were introduced into the *dcl2 dcl4* double mutant and the *dcl2 dcl3 dcl4* triple mutant. *FAD2* cosuppression was mostly released in the *dcl2 dcl4* double mutant (Figure 1A,E,F), revealing that DCL2 and DCL4 work redundantly in *FAD2* cosuppression. However, the cosuppression frequency in the *dcl2 dcl3 dcl4* triple mutant background was higher than that in the *dcl2 dcl4* double mutant (Figure 1A,E,F and Appendix A). This result was consistent with the fact that the frequency of cosuppression in the *dcl3-1* background alone was higher than that in the WT. Taken together, these results demonstrate that DCL2 and DCL4 function redundantly and counteract DCL3 in *FAD2* cosuppression.

Given that the strong cosuppression of *NIA2* was released completely by crossing with the *ago1-27* mutant [30], we transformed the *FAD2*, also driven by phaseolin promoter, into a mutant of *AGO1* (*ago1-25*) and examined the fatty acid phenotype of the T_1_ seeds. The result showed that the ratios of (18:2 + 18:3)/18:1 increased in most Pha::*FAD2*/*ago1-25* T_1_ seeds (Figure 1G), indicating that *FAD2* cosuppression was released in the *ago1* mutant and, equally, that AGO1 is involved in strong *FAD2* cosuppression.

### 2.2. FAD2 Cosuppression Is Enhanced by Mutations in RNA Decay Pathways

The dysfunction of both XRN4/EIN5 and SKI2 causes extreme phenotypes due to the silencing of endogenous genes mediated by RDR6 [20]. To determine the role of RNA decay in *FAD2* cosuppression, the Pha::*FAD2* construct was introduced into each of the *ein5-1*, *ski2-3*, and *rdr6* single mutants and the *rdr6 ein5 ski2* triple mutant. Fatty acid profiles were determined in the T_1_ seeds of the *FAD2* transgenes. While there were some exceptions (“outliers”) where cosuppression was not observed in the WT background (Figure 1A), it is worth noting that, equally, there were some outliers in the *ein5-1* (Figure 2A) and *ski2-3* (Figure 2B) backgrounds. The percentages of outliers and the ratios of (18:2 + 18:3)/18:1 in these lines also resemble those in the WT (Figure 1A,B and Appendix A). These results may suggest that unsilenced events are not affected by the related RNA decay system. However, nearly all silenced lines in the *ein5-1* and *ski2-3* backgrounds showed similar and more potent decreases in the (18:2 + 18:3)/18:1 ratios (Figure 2A and Figure 2B, respectively) compared with the silenced lines of Pha::*FAD2*/Col-0 (Figure 1A). These results indicated that the degree of *FAD2* cosuppression was enhanced in *ein5-1* and *ski2-3*.

Moreover, almost all of the T_1_ seeds in the *rdr6 ein5 ski2* harboring the *FAD2* transgene exhibited a decrease in the (18:2 + 18:3)/18:1 ratio, although to a weaker extent (Figure 2D). This phenotype was further confirmed in the T_2_ seeds of Pha::*FAD2*/*rdr6 ein5 ski2* transgenic lines, and 18:1 increased in 29 of 33 lines (Figure 2E). In order to test whether *FAD2* was still suppressed in the lines with increased 18:1, the expression of *FAD2* was determined in the T_2_ lines with different (18:2 + 18:3)/18:1 ratios (Appendix A). The expression of both endogenous and total *FAD2* decreased in lines with a low (18:2 + 18:3)/18:1 ratio, while the expression of total *FAD2* increased in the line with a high (18:2 + 18:3)/18:1 ratio (Figure 2F). These results show that although less potent (lower than the average suppression indicated by the dashed lines in Figure 2E), *FAD2* cosuppression does occur in the *rdr6 ein5 ski2* triple mutant. Since *FAD2* cosuppression was not observed in any of the Pha::*FAD2*/*rdr6* T_1_ seeds (Figure 2C), the relatively weaker cosuppression observed in *rdr6 ein5 ski2* indicates that other RDR6-independent mechanisms may be activated or enhanced by the impairment of the RNA quality control system in *rdr6 ein5 ski2*.

## 3. Discussion

### 3.1. FAD2 Cosuppression Provides a New Model for the Quantitative Study of RNA Silencing

Combined with the use of a visible marker to easily and rapidly identify transgene-positive T_1_ seed [31] and remove segregated wild-type seeds from the T_2_ population, there are a number of advantages in using *FAD2* cosuppression as a model to study the phenomenon of cosuppression. These include: (1) The expression level of *FAD2* is high in seeds and it is strongly correlated with polyunsaturated fatty acid (PUFA) levels seed oil [27]. The fatty acid profile is stable and easily quantified with samples as small as a single *Arabidopsis* seed [32]. Our extensive bench experience has indicated that seed fatty acid profiling is much more stable and reproducible than measuring gene expression levels. (2) In contrast to the phenotypic variation in *CHS* from flower to flower and petal to petal [3] and in the chlorophyll content changes from leaf to leaf within the same transgenic plant [8], there is very little variation in the T_2_ fatty acid content in our measurements. Furthermore, our results demonstrate that the fatty acid profile of each T_1_ seed represents the phenotype of an independent transformation event. (3) The frequency of *FAD2* cosuppression is as high as 80% and its intensity is as strong as the phenotype of a *FAD2* knockout mutant [27]. Cosuppression intensity and frequency can be quantified rapidly and on a large scale, with high statistical confidence, especially when T_1_ seeds are used. This quantification of cosuppression made it possible to shed light on the existence of an RDR6-independent silencing mechanism (discussed below). (4) Compared with the identification of silencing-related genetic factors with crossing as in previous studies [33], direct transformation yields a result in a shorter period of time, especially when double (e.g., *dcl2 dcl4* in Figure 1E) or triple (e.g., *rdr6 ein5 ski2* in Figure 2D) mutants are tested as host background material. (5) Seed fatty acid phenotypes are variable in T_3_ sister lines from each independent line of *B*. *carinata* [27], and this transgenerational instability behavior is quite common in transgenic activities [3]. *FAD2* transgenic lines may be a robust model for studying why this is so. (6) RNA processing, including RNA modification and splicing, has a crucial effect on RNA silencing [34,35]. Considering the possible embryo lethal of *Arabidopsis* mutants lacking key factors in RNA modification and splicing, the reverse genetic technique may be more efficient to screen the mediator of RNA silencing from the key factors of RNA modification and splicing.

### 3.2. FAD2 Cosuppression Shares Similar Genetic Factors with Other Instances of Transgenic Sense Gene Silencing and Endogenous Gene Silencing and also Exhibits an RDR6-Independent Component

To identify some of the genetic factors involved in *FAD2* cosuppression, the seed-specific *FAD2* over-expression vectors were transformed into *Arabidopsis* mutant backgrounds, wherein possible candidate factors were deleted. Frequencies and intensities were evaluated by phenotyping the fatty acid content of transgenic seeds. DCL2 and DCL4 have been identified as functioning redundantly to dice dsRNA in AGO1 cosuppression, triggered by miR168 [7]. To our knowledge, this is the only report in which DCLs were found to be involved in strong cosuppression. To identify the role of DCLs in *FAD2* cosuppression, *FAD2* was transformed into mutants of *dcl2*, *dcl3,* and *dcl4* and their corresponding double or triple mutants. Release from *FAD2* cosuppression is near-complete in the *dcl2 dcl4* double mutant, but not in either single mutant (Figure 1A,E,F and Appendix A), which suggests that DCL2 and DCL4 function redundantly in *FAD2* cosuppression. *FAD2* cosuppression was mildly relieved in the *dcl2* mutant background but not in *dcl4* (Figure 1A,B,D and Appendix A), which is consistent with the fact that 22nt siRNA is more efficient than 21nt siRNA in the PTGS pathway [33,36,37]. It is noteworthy that the absence of DCL3 leads to a higher frequency of cosuppression in both Col-0 and *dcl2 dcl4* backgrounds (Figure 1 and Appendix A), which is consistent with the results observed in the case of *AGO1* cosuppression [7]. Furthermore, the average ratio of 18:2 + 18:3/18:1 is higher in *dcl2 dcl4* than in *dcl2 dcl3 dcl4* (Figure 1A,E,F and Appendix A). These results suggest that DCL3 may function in an antagonistic way to DCL2 and DCL4 in *FAD2* cosuppression. Additionally, *FAD2* cosuppression was released in *ago1-25* (Figure 2F), which is consistent with the result observed with respect to *NIA2* [30], collectively suggesting that common genetic factors may be involved in the downstream pathway of transgenic gene silencing.

The RNA decay system mediated by XRNs and SKIs controls RNA quality. Aberrant RNA from transcription and processing can be degraded from 5′ to 3′ by XRNs [38,39,40] and from 3′ to 5′ by SKIs [41,42]. XRN4/EIN5 and SKI2 are cytoplasmic proteins, and the mutations of XRN4/EIN5 and SKI2 cause strong endogenous gene silencing and striking phenotypes in *Arabidopsis* [20]. In previous transgenic studies, the phenotype of gene silencing was enhanced by the *xrn4*/*ein5* mutation [17,22,43]. SKI2 was reported to degrade 5′ end fragments from microRNA-targeted RNAs [44]. In this study, *FAD2* cosuppression was greatly enhanced in both *ein5-1* and *ski2-3* mutant backgrounds, suggesting that both 5′- and 3′-end aberrant RNAs are involved in the process of *FAD2* cosuppression. However, in contrast to most PTGS of endogenous genes in *ein5-1 ski2-3*, which are removed by an *rdr6* mutation [20], *FAD2* cosuppression could not be effectively released in the *rdr6 ein5 ski2* triple mutant, and *FAD2* expression was suppressed (Figure 2D–F and Appendix A). Combined with the release of *FAD2* cosuppression in the *rdr6* single mutant, it appears that RDR6 plays the main role in the case of *FAD2* cosuppression, and there is a minor RDR6-independent bypass of gene silencing, which is activated or enhanced only under dysfunction of the RNA decay system. Further investigation of *FAD2* cosuppression in the *rdr6 ein5 ski2* triple mutant may uncover new genetic factor(s) or gene silencing pathways involved in cosuppression.

In summary, identifying genetic factors involved in *FAD2* cosuppression may contribute to a full understanding of the mechanism of gene silencing, which may be also helpful for designing strategies to, for example, increase PUFAs in seed oils. The frequency and intensity of *FAD2* cosuppression can be easily and quantitatively determined, making it a powerful model for studying the mechanism of cosuppression, especially cosuppression in seeds. Genetic factors, as well as an RDR6-independent mechanism, were unveiled by quantitatively investigating this strong cosuppression model (Appendix A). *FAD2* strong cosuppression shares some common genetic factors associated with low-frequency transgenic PTGS, endogenous PTGS, and microRNA gene silencing [7,11,12,13,30], suggesting that the differences among various examples of transgenic gene silencing may be due to the initiation of silencing. Additional studies will be important to further elucidate the unknown factor(s) involved in the RDR6-independent pathway and the initiation mechanism of cosuppression. 

In plant genetic engineering, the expression of transgenes often causes different degrees and frequencies of gene silencing [11]. The gene silencing pathway is involved in various metabolic pathways necessary for plant growth, development, and stress resistance. Thus, knocking out the key elements of gene silencing could avoid silencing caused by transgenes but also cause serious negative effects on plants [12,30], which lead to no application value of transgenic lines. Therefore, elucidating the initiation mechanism of gene silencing is essential for designing a strategy to prevent the low expression efficiency of transgenes of target phenotypes in plant bioengineering.

## 4. Materials and Methods

### 4.1. Plant Materials and Growth Conditions

Seed germination and plant growth for *Arabidopsis* were previously described [27]. All the *Arabidopsis* mutants used in this study were of the Columbia (Col-0) background. *dcl2-1* (*N16389*), *dcl3-1* (*N505512*), *dcl4-2T* (*N66075*), and *dcl2-1 dcl4-2* (*N66078*) single mutants and the *dcl2-1 dcl3-1 dcl4-2* (*N16391*) triple mutant were obtained from the Nottingham *Arabidopsis* Stock Centre. Mutants *rdr6-11*, *ein5-1*, *ski2-3* (*Salk_063541*), and *rdr6-11 ein5-1 ski2-3* were kindly provided by Hongwei Guo [20]. Mutant *ago1-25* was a gift from Yijun Qi [45]. A PCR was performed to genotype these mutants, and the PCR primers used for genotyping are listed in Table S3. Seeds were surface sterilized, plated on 1/2 MS medium with 1.5% sucrose, and stratified at 4 °C for 3 d. Then, one-week-old seedlings were planted in the soil mixture (enriched soil: vermiculite: perlite = 3:1:1, V:V:V) in a growth chamber with 16 h of light (200 μmol·m^−2^·s^−1^ radiation) and 8 h of dark at 22 °C.

### 4.2. Vector Construction and Plant Transformation

The vector carrying phaseolin-driven *AtFAD2*, pK7WG2D-Pha-*FAD2*-DsRed (Pha::*AtFAD2*), was constructed as described previously [27]. Phaseolin is a seed-specific promoter. There is also a *DsRed* expression cassette on the Pha::*AtFAD2* vector to facilitate the screening of transgenic-positive seeds. *Arabidopsis* transformation was performed using the Agrobacterium-mediated floral dip method [46]. 

### 4.3. Fatty Acid Analysis of T_1_ Single Seeds 

The transgenic-positive T_1_ seeds were screened with red fluorescent DsRed. Then, the fatty acid composition of the T_1_ seeds was determined using gas chromatography (GC) with a single seed as a biological replicate, as described previously [47]. Firstly, a single seed was put into a gas chromatography sample vial (GC-vial) with 200 μL transmethylation solvent (5% sulfuric acid methanol with 30% toluene) and incubated in an 85 °C water bath for 2 h. The samples were removed from the water bath and cooled to room temperature, and then 200 μL 0.9% NaCl solution (wt/wt) was added to terminate the reaction and 100 μL n-hexane was added to extract the fatty acid methyl esters (FAMEs). This was followed by vortex for 30 s and centrifugation at 1500× *g* for 10 min. Lastly, 60 μL FAMEs-hexane solution was transferred into an insert in a GC vial for the GC assay. The fatty acid compositions were analyzed according to the retention time and peak area. The fatty acid compositions of the T_2_ generation were determined with 20 seeds as a biological replicate and the transmethylation system was doubled.

### 4.4. RNA Extraction and Gene Expression Analysis 

The total RNA of the developing seeds at 12–14 days after flowering was extracted with a Rapid Extraction Kit (RP3202, Bioteke Corporation, Wuxi, China). Then, a genome-eraser kit (PrimeScript^TM^ RT reagent Kit with gDNA Eraser, Takara Cat # RR047A, Dalian, China) was used to obtain the cDNA for the following gene expression analysis. To analyze the relative expression level of total and endogenous *FAD2*, qRT-PCR was performed on the ABI QuantStudio 7 Flex Real-Time PCR system (Thermo Fisher Scientific, Waltham, MA, USA). The primers Endo-FAD2-F (CTTCTTCTTCGTAGGGTG) and Endo-FAD2-R (TGTTTCTGGAGATGGAGC) located in the 5’-UTR of *FAD2* were designed to determine the relative expression level of endogenous *FAD2*, while the primers total-FAD2-F (GCTGGATGACACAGTTGGTCTTATCTT) and total-FAD2-R (GGAATGGTGACGGCGATGACTATAC), located in the *FAD2* coding region, were designed to determine the relative expression level of total *FAD2*. The *Tubulin* gene, designed with the primers Tub-F (TTTGTGCTCATCTTGCCACGGAAC) and Tub-R (TTTGTGCTCATCTTGCCACGGAAC), was used as a reference gene. The relative expression level was calculated using the 2^−ΔΔCT^ method.

## Figures and Tables

**Figure 1 ijms-24-17165-f001:**
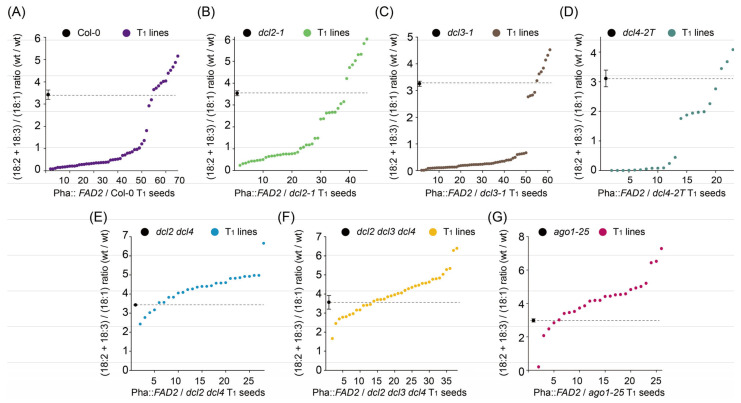
Intensities of *FAD2* gene silencing in mutants of DCLs and AGO1. This figure shows the ratio of (18:2 + 18:3)/18:1 proportions in T_1_ seeds expressing Pha::*FAD2* in Col-0 (**A**), *dcl2-1* (**B**), *dcl3-1* (**C**), *dcl4-2T* (**D**), *dcl2 dcl4* double mutant (**E**), *dcl2 dcl3 dcl4* triple mutant (**F**), and *ago1-25* (**G**). Pha::*FAD2* indicates that *FAD2* is driven by the *Phaseolin* promoter. DCL refers to the dicer-like protein and AGO refers to AGONAUTE. T_1_ seeds are identified by the red fluorescence from transformed plants. The values 18:2 and 18:3 represent linoleic acid and linolenic acid, which are the product of FAD2, and 18:1 represents oleic acid, which is the substrate of FAD2. The (18:2 + 18:3)/18:1 ratio is calculated using the weight percentages of the mentioned fatty acids and represents the expression of *FAD2*. The values of the (18:2 + 18:3)/18:1 ratio in wild-type Col-0 seeds and *dcl* mutant seeds show the average ratio ± SD (bar, *n* = 3). The dashed lines are used to compare the average ratios in wild-type seeds or those of the corresponding untransformed mutant. See also Appendix A.

**Figure 2 ijms-24-17165-f002:**
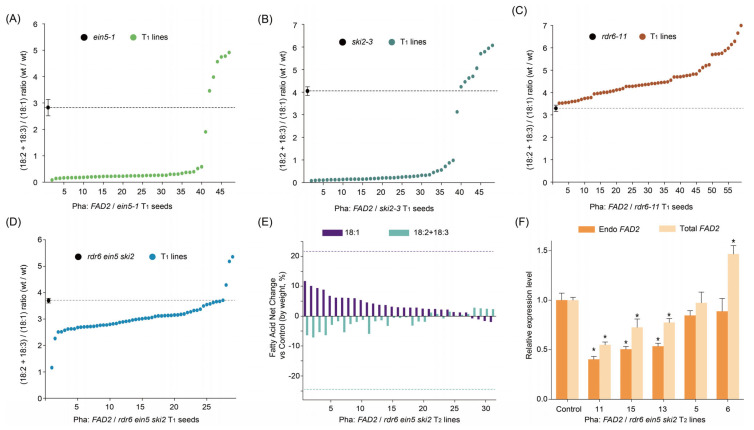
Intensities of *FAD2* gene silencing in mutants of the RNA decay system and *rdr6*. This figrue shows the ratio of (18:2 + 18:3)/18:1 proportions in the T_1_ seeds expressing Pha::*FAD2* in *ein5-1* (**A**), *ski2-3* (**B**), *rdr6-11* (**C**), and the *rdr6 ein5 ski2* triple mutant (**D**). The values 18:2 and 18:3 represent linoleic acid and linolenic acid, which are the product of FAD2, and 18:1 represents oleic acid, which is the substrate of FAD2. The (18:2 + 18:3)/18:1 ratio is calculated using the weight percentages of the mentioned fatty acids and represents the expression of *FAD2*. (**E**) shows net changes in proportions of 18-carbon fatty acids in T_2_ seeds of Pha::*FAD2*/*rdr6 ein5 ski2* lines. (**F**) shows the relative expression levels of total *FAD2* and endogenous *FAD2* (Endo *FAD2*) in 12–14 day developing seeds from representative lines of Pha::*FAD2*/*rdr6 ein5 ski2*. EIN5: ethylene-insensitive 5; SKI2: Super-Killer 2; RDR6: RNA-dependent RNA polymerase 6; AGO1: ARGONAUTE 1; Endo FAD2: endogenous FAD2. Pha::*FAD2*/mutant refers to *FAD2*, driven by the *Phaseolin* promoter, transferred to the mutant host. T_1_ seeds were selected from transformed plants exhibiting red fluorescence. The values of mutant seeds show the average ratio ± SD (bar, *n* = 3). The dashed lines are used to compare the average ratio in seeds of corresponding untransformed mutants (**A**–**D**). Ten positive T_2_ seeds (exhibiting red fluorescence) were used for fatty acid analysis. The dashed lines show 18:1 and 18:2 + 18:3 changes in the *fad2-1* mutant vs wild type, respectively. In (**E**), Endo *FAD2* was detected using primers in the 5’-UTR of *FAD2*, while total *FAD2* was detected using primers in the coding region. The relative expression levels show the average ratio compared with the untransformed *rdr6 ein5 ski2* mutant ± SD (bar, *n* = 3) (**F**). An asterisk (*) indicates a significant difference (*t*-test, *p* < 0.05). See also Appendix A.

## Data Availability

Data are contained within the article.

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

*Arabidopsis* and camelina seeds. Plant Direct.

