# Peer review of "Quantitative Investigation of FAD2 Cosuppression Reveals RDR6-Dependent and RDR6-Independent Gene Silencing Pathways"

_ijms, 2023, doi:10.3390/ijms242417165_

Round 1

Reviewer 1 Report

Comments and Suggestions for Authors

Review Manuscript (ijms-2725028) “Quantitative investigation of FAD2 cosuppression reveals 2 RDR6-dependent and RDR6-independent gene silencing path-3 ways” by Chen et al. presents and interesting overview of gene silencing study in Arabidopsis. Theses results could be of great interest for plant physiologists.

However, the defects in the form and the style, and in the description of protocol and exposition and discussion of results made this manuscript unacceptable for the publication in International Journal of Molecular Science. In general, experimental design is very deficient and the exposed results very poor.

Most important defects for the rejection of manuscript are:

In the whole manuscript all acronyms indicated in the text must be described during the first citation.

Around the whole manuscript names of genes must be indicated in italics.

Objectives of the review must be clarified in a separated paragraph on the Introduction section witout references.

Information of Figure 1 is not clear regarding the explanation in the legen. In addition, regarding the experimental design the analysis performed are not clear regarding the assayed comparisons.

Information of Figure 2 is not clear. Any explanation of the treatments is indicated. The bioinformatic analysis is not included in the Material and Method sections. For example, expression “Ratio of (18:2 + 18:3)/18:1 proportions” is not described in the description of the experimental design.

Legend of all Figures must be clarified. All acronyms indicated in the figured must be clarified in the legend. These figures must be self-readable.

Material and Methods section is very poor. Arabidopsis clones must be better described. Plant transformation protocols are very poor. qPCR analysis is also poor, any indication of the biological and technical replications is included, no statistical analysis neither a clear experimental design.

In addition, a final Conclusion paragraph introducing the main implications for plant production and breeding of the obtained review results.

Comments on the Quality of English Language

English grammar and expression should be revised

Reviewer 2 Report

Comments and Suggestions for Authors

The authors investigate the mechanisms leading to FAD2 cosuppression.

It is an interesting study.

I have one major remark. I am a specialist of plant lipids, but not of genetic regulation.

So for me I think sometimes a clear presentation of the investigated mechanism lacks.

In the abstract you write “In the previous studies, we show that expression of FAD2 in the seeds of Arabidopsis results in RDR6-mediated strong cosuppression.”

May be one sentence after this one you could explain what is co suprression: what is “co supressed”? what   are the genes that are co suppressed?

Co suppression and co inhibition, is it the same? In the Abstract we find, “co suppression” , “co repression” and “co inhibition”? are all these terms equivalent? Is not it confusing to use 3 terms for the same phenomenon. You need to define more clearly what you mean by these terms.

Line 25: why “second”?

Naïve question; is the rate of transcription of the transgene important for the cosupression phenomenon?

Line 72: italics for (Linum usitatissimum)

The overexpression was driven by what promoter?

Line 88: is it the same promotor than the one used in the previous study?

Figure 1 is not that clear. Do you do this study seed by seed? Each value is the result of one seed??

I guess you want to show the heterogeneity of the phenotype, which is good. Would not it be nice to also show the distribution of the phenotype

For 1F, 1G and 1E should not you have a “mean” (since the distribution seems to be normal)

Why in 2C, all the T1 are higher than in the rdG-11 control? It is not suppression but overexpression, right?

In Materials and Methods I see nothing about fatty acid analysis…

But I see things about RNA

Have you done RNA analysis? Where are they?

By the way, you speak of co suppression; you need to explain why the fatty acid reflects the gene expression. I guess you showed it in the previous paper but here also we need a correlation between RNA level and FA in seeds

Reviewer 3 Report

Comments and Suggestions for Authors

Although there are several differences between plant genes in how often and how much transgene-mediated cosuppression happens, not much research is known regarding underlying processes that cause strong cosuppression.

The researchers have shown FAD2 is expressed in Arabidopsis seeds and leads to strong cosuppression through RDR6. In this authors studied the FAD2 strong co-inhibition system for identification genetic factors that affect the production of FAD2 in mutants. And the factors such as DCL2, DCL4, AGO1, and EIN5 were first found to be involved in FAD2 cosuppression. It

Overall, the study is well performed and presented.

Some minor comments:

1.     Authors should provide high quality images/figures, currently the figures resolution is very poor.

2.     Section 5 on patents is unnecessary.

3.     Figure caption needs to be elaborated.

4.     Authors could consider to briefly discuss impact of RNA methylation (see PMID: 36938064 or PMID: 37767300) on gene silencing in plants (one or two sentences) in introduction or discussion.

Comments on the Quality of English Language

Minor editing is requested .

Round 2

Reviewer 1 Report

Comments and Suggestions for Authors

Authors have revised correctly the manuscript

Reviewer 2 Report

Comments and Suggestions for Authors

The authors thoroughly responded to my concerns

nice job